How do colonial Eurasian Griffon Vultures prevent extra-pair mating?

Bertran Joan 1
Macià Francesc Xavier 1
Margalida Antoni 2 3 amargalida@prodan.udl.cat
1 Bearded Vulture Study and Protection Group , El Pont de Suert , Spain
2 Department of Animal Science (Division of Wildlife), Faculty of Life Sciences and Engineering, University of Lleida , Lleida , Spain
3 Division of Conservation Biology, Institute of Ecology and Evolution, University of Bern , Bern , Switzerland
Wink Michael
Electronic publication date: 2016 Feb 29
Publication date: 2016
Volume: 4
Electronic Location ID: e1749
Received 2015 Nov 19; Accepted 2016 Feb 10
Copyright: ©2016 Bertran et al.
Copyright year: 2016
Copyright holder: Bertran et al.
License: This is an open access article distributed under the terms of the Creative Commons Attribution License, which permits unrestricted use, distribution, reproduction and adaptation in any medium and for any purpose provided that it is properly attributed. For attribution, the original author(s), title, publication source (PeerJ) and either DOI or URL of the article must be cited.
License URL: https://creativecommons.org/licenses/by/4.0/

Keywords: Copulations, Extra-pair mating, Colonial species, Vulture, Sperm competition

Funding: Ministry of Economy and Competitiveness RYC-2012-11867 AM was supported by a Ramón y Cajal research contract from the Ministry of Economy and Competitiveness (RYC-2012-11867). The funders had no role in study design, data collection and analysis, decision to publish, or preparation of the manuscript.

==============================
In colonial breeding species, preventive measures to reduce the risks of extra-pair copulations (EPCs) should reflect the actual risk perceived by males (e.g., proximity of neighbors, intrusions into the nest) mainly during the fertile period. In colonial vultures, specific studies examining the preventive measures that minimize the risks of EPCs occurring within the competitive context of colonial breeding have not been conducted. Here we tested at Eurasian Griffon Vulture (Gyps fulvus) nesting sites the intensity of paternity assurance behavior, shown as frequency and duration of within-pair copulations (WPCs), potential mate vigilance or nest attendance, and levels of aggressivity. This was measured according to the frequency of territorial intrusions and comparison of the fertile vs. the non-fertile period. Our findings suggest that the frequency of WPCs and their duration increased significantly during the presumed fertile period, regarded as the period when Griffon pairs spent significantly more time together at their nests. In addition, low levels of territorial intrusions were observed, an aggressive response of pairs towards intruders, and a relatively high presence of pairs at the nests during the fertile period. Thus, although nesting sites are subject to low exposure to EPC attempts, the increased frequency and duration of copulations during the fertile period suggests that, under pressure from the colonial breeding system, a higher rate of copulations is the most effective preventive mechanism against relative uncertainty of paternity.

Introduction

For colonially breeding bird species, competition for mates is one of the inevitable costs associated with reproduction. This may be a consequence of socio-ecological factors that make male mate-guarding an insufficient measure, or of the close proximity of neighbors facilitating situations favorable to sperm competition (Wittenberger & Hunt, 1985; Møller & Birkhead, 1993; but see Westneat & Sherman, 1997).

In raptors, proximity of conspecific breeding sites is associated with an increased risk of extra-pair copulations (EPCs), and consequently the intensity of preventive mechanisms increases with breeding density (Simmons, 1990; Arroyo, 1999; Mougeot, Arroyo & Bretagnolle, 2001; Mougeot, 2004). Paternity assurance strategies in solitary raptors, as seen in other colonial species, include (i) frequent within-pair copulations (WPCs), (ii) guarding of the partner in the nest as in the traditional mate guarding of solitary bird species, and (iii) intraspecific aggression (Negro, Donázar & Hiraldo, 1992; Arroyo, 1999; Mougeot, Arroyo & Bretagnolle, 2001; García & Arroyo, 2002; Mougeot, Arroyo & Bretagnolle, 2006). However, paternity assurance behavior in colonial breeding species can be costly in terms of time and energy because males have to divide the time between remaining vigilant near the nest and visiting foraging sites (Birkhead & Møller, 1992; Møller & Birkhead, 1993). This would be especially relevant when food resources are temporarily scarce and scattered or distant from nesting sites. Consequently, the preventive measures to reduce the risks of EPCs should reflect the actual risk as perceived by males.

In colonial raptors, as well as aquatic and seabird species, copulation attempts away from nesting sites are unusual (Negro & Grande, 2001). In the Eurasian Griffon Vulture (Gyps fulvus), as well as other colonial vultures (Robertson, 1986), pre-laying courtship takes place mainly on nesting sites (Xirouchakis & Mylonas, 2007; Margalida & Bertran, 2010), where pairs remain most of their time. However, in this species and other colonial vultures EPCs have been recorded on nesting sites (Robertson, 1986; Xirouchakis & Mylonas, 2007). Several studies have found that breeding density and extra-pair paternity rate are positively correlated in populations of the same species (Møller & Ninni, 1998). Thus, in dense colonies EPCs should be assessed as scenarios potentially advantageous in sexual competition through matings (Mougeot, 2000).

The Eurasian Griffon Vulture is a cliff-nesting, socially monogamous, colonial species that may breed in large colonies (del Hoyo, Elliott & Sargatal, 1994).This species exhibits relatively high copulation rates over an extended period (Margalida & Bertran, 2010). Most raptors copulate extensively before egg-laying and consequently a number of copulations take place outside the fertile period (Negro & Grande, 2001). Copulations outside the fertile period probably have functions related to pair bonding, mate assessment, and territorial signaling (Newton, 1979; Tortosa & Redondo, 1992; Negro & Grande, 2001). At high breeding densities, males copulate frequently when the perceived high EPC risk is mainly derived from territorial intrusions (Møller & Birkhead, 1993). Intrusions may allow floaters to exploit EPC opportunities and to increase breeding success (Cooper et al., 2009; Moulton, Linz & Bleier, 2013), although limited by the aggressive behavior of territorial males (Moulton, Linz & Bleier, 2013). In strictly colonial vultures there are no specific studies examining mating behavior (copulations, extra-pair encounters, or aggressive interactions) and the preventive behaviors that minimize EPC risk. In this sense, territorial intrusions by conspecifics generally occur irregularly and are short and hidden events, being difficult to predict and observe, even in species with very high rates of extra-pair paternity (EPP; Dixon et al., 1994; Hoi, Kristofík & Darolová, 2013). For this reason, observational data on male and female behavior and intrusions during the fertile period are scarce (Hoi, Kristofík & Darolová, 2013).

The present case study is a contribution to reducing this deficit in our knowledge by studying mating behavior in Eurasian Griffon Vultures in a Pyrenean population (NE Spain). Our prediction is that in situations of colonial nesting, the risks of territorial intrusions and extra-pair encounters should be higher during the female fertile period and, accordingly, males should consistently show an increase in preventive behavior in that period. We tested this prediction on nesting sites by examining the frequency and potential intentionality of territorial intrusions, as well as the intensity of paternity assurance behavior and the frequency and duration of WPCs, mate vigilance, nest attendance, or levels of aggressivity. These parameters were recorded and compared in the presumed female fertile and non-fertile periods.

Material and Methods

Ethics statement

All procedures regarding observational field study (Ref. 4925-2009/2011) were conducted according to the relevant Spanish legislation and following the conditions and guidelines approved by the Department of Agriculture, Livestock and Fisheries of the Government of Catalonia. It was not possible to record data blind because our study involved focal animals in the field.

Study species and study area

The Eurasian Griffon Vulture can nest in large colonies of over 150 pairs. Socially monogamous, both male and female provide long and extensive parental care for a single egg and chick (del Hoyo, Elliott & Sargatal, 1994; Xirouchakis & Mylonas, 2007). On average, the species allocated 7.6 h/day to food searching, being recorded the shortest foraging time in December (6.4 h/day) and the longest in June (9.3 h/day) (Xirouchakis & Andreou, 2009).

In breeding colonies, the distance between neighbors can be only a few meters and breeding pairs only defend the immediate vicinity of the nest. Sexual activity in the species on nesting sites began on average 84 days before egg-laying (Xirouchakis & Mylonas, 2007). Copulations have an average duration of 48–64.6 s (see Xirouchakis & Mylonas, 2007; Margalida & Bertran, 2010); they are conspicuous and accompanied by loud cries (Margalida & Bertran, 2010). In this and in similar species, copulations at the nest site probably have functions other than fertilization, such as territorial signaling (see Robertson, 1986; Negro & Grande, 2001).

We conducted fieldwork during the breeding seasons (pre-laying periods) of 2008–2011 in the Catalonian foothills of the Pyrenees (NE Spain), in six colonies with 33 breeding pairs (Table 1). The size of the colonies (maximum distance from one side to the other of the colonies) ranged approximately between 50 and 175 m. In Catalonia the breeding population in 2009 was estimated at 1,115 breeding pairs. The high densities of avian scavengers in Catalonia are a result of the extensive livestock populations (Margalida, García & Cortés-Avizanda, 2007).

Table 1 Descriptive data of the colony size, numbers of pairs controlled and fieldwork invested during the study of sexual activity on Eurasian Griffon Vultures.

Colony	Pairs controlled	Days	Hours	Colony size	
1	5	11	66.5	16	
2	5	10	60	29	
3	8	9	45.25	40	
4	5	10	43.5	19	
5	5	9	44	9	
6	5	9	83.25	8	
Total	33	57	342.5		

Data collection and observation methods

In all years, observations began during the first week of November, coinciding with the period previous to the first copulation attempts (Margalida & Bertran, 2010) and concluded with the laying period (January–February). In each colony we carried out a simultaneous monitoring of five pairs, except in one colony in which nests were very close to each other, enabling us to follow a total of eight pairs. We carried out a weekly visit per colony, thus conducting a total of 342.50 h of observations in 59 fieldwork days (average per colony 57.1 h, range 43.5–83.25 h).

In each colony the criteria used to select breeding pairs to be monitored were established by taking into account the higher nest concentrations and focal nests that had optimal observation conditions. In all cases, the distances were <100 m and only in a case the largest distance was approximately around 200 m. When possible, we also recorded the birds’ activities away from the nests (e.g., material collecting for the nests). Birds were observed with a 20–60 x telescope at a distance of 100–200 m from the rock face where the nests were located.

The Griffon Vultures in this study were not marked individually, though this species shows virtually no sexual dimorphism (del Hoyo, Elliott & Sargatal, 1994). Consequently, to identify nesting pairs and to avoid the risk of counting copulations that were possible cases of EPCs by intruders, we relied on observations of typical behavior of the pairs involved in copulations, as well as contributions to nest-material gathering, arrangement of the nest, or nest defense (see Ferrero, Grande & Negro, 2003; Margalida & Bertran, 2010). Individual characteristics of their plumage (perched and in flight) were also used to identify the partners (see Bertran & Margalida, 1999; Margalida & Bertran, 2000a). In addition, the Eurasian Griffon Vulture is a territorial species, intruders being aggressively expelled from the immediate nest surroundings during the fertile period (Xirouchakis & Mylonas, 2007; see also Results). We considered that such intrusions would occur furtively and be brief in duration, since copulations in these vultures are conspicuous because of their long duration and being marked with loud cries (Margalida & Bertran, 2010). To minimize possible replication in the copulation frequencies, we took into account the time when birds remained together before or after intra-pair matings. In this way, 74% (n = 171) of sexual interactions observed were accompanied by habitual behavior of pairs at nesting sites (i.e., nest-building, delivering material, arranging the nest, nest defense), and so we were able to rule out those cases involving foreign individuals. In the remaining 26% (n = 60), matings were “neutral” without any behavior that could be associated with the resident pairs. Thus of the 60 cases of “neutral” mountings, we discarded those in which the individuals remained together (until one left the nest) for more than 30 min, which left 16 interactions which showed an average presence of birds together of 19.9 min (range: 14–28 min). In all these cases copulations ended with apparent success and behaviors that in any way indicated the existence of EPCs were not observed (see Results). On the contrary, the EPC attempts observed during a no systematic monitoring outside nesting sites (20–200 m) could be partially underestimated. However, these EPC attempts were confirmed because when males abandoned nests, their partners remained in them, and then males interacted with other females. In this sense, except in one colony, the observations were carried out by two observers. Therefore, events in and out the nest were followed simultaneously.

During observations of focal nests, for each observed copulation attempt we recorded: (1) the identity of the pair involved; (2) whether the copulation was successful or not (i.e., whether cloacal contact was achieved during mount); (3) the duration of copulation attempts (in seconds) measured with a stopwatch, and all included mounting movements. The frequency of copulations was estimated as the number of attempts per hour. We quantified for each pair/colony the percentage of time spent by one or both members of the pair within the breeding territory (nest site and nearby area). We recorded the frequency of territorial intrusions (number of events/h) and when these occurred at nests where one or both members of the pair were present. We also recorded for each intrusion if it ended with aggression and the levels of aggressivity (i.e., physical contact or not).

The frequencies obtained for copulation behavior, nest attendance, and territorial intrusions were combined in two differentiated time periods, pre-fertile (PF) and fertile (F). Studies on copulatory behavior in raptors suggest that the fertile period can begin about 12 days before egg-laying (Bird & Buckland, 1976; Negro, Donázar & Hiraldo, 1992; Mougeot, 2000). Here, we assumed a presumed fertile period of <14 days before laying, dating backwards from egg-laying (day 0 was considered as the egg-laying date). Egg-laying dates were determined by direct observation of adult behavior in the nest.

Data analysis

All of the statistical analyses were carried out to a significance level of 0.05. The differences in the amount of time spent by males and females at the nest, intrusions, copulation attempts, and copulation duration between periods (PF vs. F) were tested using the Wilcoxon test for matched pairs. Values presented are the mean ± standard deviation (SD).

Results

Within-pair copulations

We observed a total of 231 sexual interactions on the nests. Copulation attempts were observed in 31 (93.9%) of the 33 pairs monitored. In 210 (90.9%) copulations attempts we were able to discern if these were successful, which 93.3% (n = 196) were. Non-successful mating attempts were caused by female reluctance.

The mean frequency of copulation attempts (attempts/hour) was significantly higher during the fertile period (PF: 0.10 ± 0.03 vs. F: 0.33 ± 0.05, n = 231; Wilcoxon test: z = 2.207, P = 0.027, Fig. 1A). In addition, the time duration of the behaviorally successful copulations (data obtained from five colonies, n = 120), were significantly more prolonged in this period (PF: 33.26 ± 4.16 vs. F: 43.96 ± 9.55; Wilcoxon test: z = 2.023, P = 0.043, Figs. 1B).

Figure 1 Differences in the copulation frequency (A) (attempts per hour ± SD) and time duration (B) (seconds ± SD) between the presumed pre-fertile and fertile periods (for details see Methods).

Nest presence

The average proportion of time that one of the members of the pair was present at the nest did not vary significantly between the two periods (PF: 23.67 ± 6.12% vs. F: 30.42 ± 5.25%; Wilcoxon test: z = 1.572, P = 0.115). On the contrary, the average presence of the two individuals together increased significantly during the fertile period (PF: 32.57 ± 9.10% vs. F: 47.67 ± 3.84%; Wilcoxon test: z = 2.201, P = 0.027; Fig. 2A).

Figure 2 Differences in the percentage of presence (A) (± SD) of breeding pairs at nests and the frequency of intrusions per hour (B) (± SD) at nest sites by foreign individuals between the presumed pre-fertile and fertile periods (for details see Methods).

Nest defense behavior

Nest intrusions were observed in 30 (90.9%) of the 33 monitored pairs; 14.4% (n = 15) of the agonistic interactions ended with physical aggression. The intrusions (n = 104) occurred irrespective of, and in similar proportions to, whether one partner of the pair was present at the nest (45.2%) or both (54.8%). Similar intrusion proportions were observed in both the pre-fertile and fertile periods (PF: partners in the nest: one 45.7%, both 54.3%, n = 70; F: one 44.1%, both 55.9% N = 34; χ12=0.584, P = 0.445). The average frequency of intrusions (intrusions/hour) was marginally significantly higher in the fertile period (PF: 0.05 ± 0.03 vs. F: 0.11 ± 0.04; Wilcoxon test: z = 1.941, P = 0.052; Fig. 2B).

In two of the six studied colonies we observed EPCs. These EPCs involved four males (12.1%) and a female (3%), being 3.3% of observed copulations (n = 239). The EPCs were effectuated by males in neutral sites in the colonies at distances of between 20 and 200 m from their nests. In four of the cases, the interactions occurred while males were absent collecting nest material. One case of successful EPC is highlighted which involved a male and a female from neighboring nests (separated by 10 m), both during their respective fertile periods, while collecting material for their nests. All extra-pair encounters were brief, and in seven of the eight cases the copulation attempts by males were rejected by the females involved.

Discussion

Within-pair copulations

Usually all raptors exhibit high rates of copulation during an extended period of time (Negro & Grande, 2001), and in some species the frequency of intra-pair copulations increases with the breeding density (Simmons, 1990; Korpimäki et al., 1996; Arroyo, 1999; Mougeot, 2004). However, there is great interspecific variation and it is not always the case that colonial species show higher copulation rates than solitary species, which disagrees with the hypothesis of sperm competition, suggesting that phylogenetic aspects should be also evaluated (Arroyo, 1999). For example, Griffon Vultures averaging 71.7 copulations per clutch and an average frequency of 1.2 copulation/day (Margalida & Bertran, 2010) show a frequency lower than the average copulatory behavior observed in other raptors (215 copulations per clutch and 11 per day; Arroyo, 1999). However, our comparative results between the different stages of the pre-laying period show that both, the relative frequency of intra-pair copulation and its duration, increased significantly during the presumed fertile period, being consistent with the hypothesis of paternity insurance/sperm competition (Birkhead & Møller, 1998; Mougeot, 2000; Komdeur, 2001; Mougeot, Arroyo & Bretagnolle, 2001; García & Arroyo, 2002). The duration of copulations is associated with an increased sperm transfer mechanism to dilute the sperm of other males in situations of sperm competition (Birkhead & Møller, 1992). There is unfortunately little information for raptors, and a longer duration of copulation could result in a greater transfer of sperm or ensure cloacal contact (Mougeot, 2004). For example, an experimental study in the semi-colonial Montagu’s Harrier (Circus pygargus) showed that males increase both WPCs and copulation duration in simulated situations of sperm competition.

Nest attendance, nest defense, and territorial intrusions

Griffon pairs spent significantly more time together in the nest in the presumed fertile period. Although raptor males are considered to be inefficient in mate guarding, in some species they seek to maximize their time with females on nesting sites during their fertile period (Birkhead & Møller, 1992). Specifically, in some territorial vultures like Egyptian Vulture (Neophron percnopterus) and Bearded Vulture (Gypaetus barbatus) (Donázar, Ceballos & Tella, 1994; Bertran & Margalida, 1999) males significantly increased their time at the nest together with females in mate vigilance behavior. This can be facilitated because in these species, as occur in Griffon Vultures, males do not feed their partners (courtship feeding) during the fertile period (Margalida & Bertran, 2000b). However, a previous study showed that male Griffons did not significantly increase their time with females as the time of egg-laying approached (Margalida & Bertran, 2010). The possible absence of mate guarding in this species, which covers large areas searching for an unpredictable and scarce food resource, may be due to a conflict arising from increasing surveillance of the nest and reduced foraging efficiency (Møller, 1987; Westneat, 1994). In fact, Griffon Vultures annually invest on average about 58–75% of their time to foraging activities (Leconte, 1977; Xirouchakis & Andreou, 2009). However, the male presence at the nest and its surroundings during the hypothetical fertile period has to be relevant because males collect most of the material for the nest, which takes place within two weeks of egg-laying. In this period we observed the 79.4 ± 14.8% of the total deliveries (range 58.8–100%, n = 258) in which in the 68.4% (n = 79) were involved the males (J Bertran, FJ Maciá & A Margalida, 2010, unpublished data; see also Xirouchakis & Mylonas, 2007).

Intrusions occurred in 90.9% of the nests monitored, with a tendency to increase (although statistically marginal) during the presumed fertile period. However the level of intrusions observed (0.05 and 0.11/h during the pre-fertile and fertile periods, respectively) is low if we hypothetically consider this colonial species to be prone to EPC attempts. For example, in some semi-colonial raptor species like Red Kite (Milvus milvus) territorial intrusions by males are relatively frequent during the fertile period (Mougeot, 2000). Intrusions by outsiders frequently occur for reasons associated with territoriality and the search for vacant sites, but in the female fertile period they can involve intrasexual competition (Møller, 1987). Our study does not show this clearly, since it would be expected that males should seek EPCs in other nests, taking advantage of this opportunity when females were alone. However, the results show that the similar numbers of intrusions occurred when the nest was occupied by one member of the pair (45.2%) or both (54.8%), and in both cases the intruders were expelled, a relatively high proportion (14.4%) of them by physical aggression. In this species, attempts to steal nest material from neighboring sites can be a major cause of intrusions. These actions (which can also occasionally include nest destruction) occur mostly when nests are unguarded, but can also happen when they are occupied by pairs, and can coincide with the female fertile period (J Bertran, FJ Maciá & A Margalida, 2010, unpublished data; see also Xirouchakis & Mylonas, 2007).

Extra-pair copulations

The frequency of copulations outside the pair bond in socially monogamous species obliges males to adopt preventive strategies to avoid the risk of cuckoldry (Birkhead & Møller, 1992; Hoi, Kristofík & Darolová, 2013). In this sense, it is difficult to disentangle when females use male absence to obtain EPCs and when they suffer male harassment and coercion in forced EPCs (Dunn et al., 1999; Low, 2004; Low, 2005).

Studies on the copulatory behavior of raptors showing the frequency of EPCs are relatively scarce: 7% (see Arroyo, 1999; Mougeot, 2004) of the 287 known species (Newton, 1979). The percentage values of EPCs found in the studied species vary between 0 and 7.3% (Mougeot, 2004; see also Arroyo, 1999). With respect to EPP (extra-pair paternity), in species such as Swainson’s Hawk (Buteo swainsoni) the frequency is low, occurring in 5% of chicks and 7% of nests (Briggs & Collopy, 2012). The percentage of EPCs obtained in our study (3.3% of copulations, n = 239) coincides with values documented in Xirouchakis & Mylonas (2007), but unlike these authors we did not confirm episodes of EPCs on nesting sites. The fact that EPC attempts were observed in sites not surveyed systematically (20–200 m from the nests) suggests the possibility that the actual frequency of EPCs was greater than obtained. EPC attempts occurred in two (33%) of the six colonies studied and involved only 12.1% of males and 3% of females monitored (n = 33 breeding pairs) and in most attempts (88%) EPCs were rejected by females.

Levels of EPCs in other vulture species are also low: 0.05% in Cape Griffon (Gyps coprotheres) (Mundy et al., 1992); 0.52% in Bearded Vulture (Bertran & Margalida, 1999); 2.6% in Egyptian Vulture (Donázar, Ceballos & Tella, 1994). But this low frequency of EPCs contrasts with the 23% obtained in a reintroduced population of California Condor (Gymnogyps californianus) (Mee et al., 2004). These authors argued that EPCs may be enhanced in this reintroduced population because of increasing social interactions due to food concentration at a few feeding stations, limited mate choice, and a high level of inbreeding (Mee et al., 2004).

Extra pair fertilizations (EPFs) as a consequence of EPCs have been detected at low levels in the raptor species studied (1–5% of young or broods; Mougeot, 2004; Rosenfield et al., 2015). Based on the idea that females control the success of copulations, we can expect that in long-lived species where males invest heavily in reproduction (e.g., colonial seabirds and raptors) they obviously tend to restrict EPCs, to avoid jeopardizing the male investment (Whittingham, Taylor & Robertson, 1992; Westneat & Sargent, 1996; Petrie & Kempenaers, 1998; Sheldon & Ellegren, 1998; Briggs & Collopy, 2012; Wojczulanis-Jakubas, Jakubas & Chastel, 2014). In this regard, in raptors that nest in open country the fact that extra-pair interactions are visible or audible at long distances has also been suggested as a limiting factor in EPCs (Korpimäki et al., 1996). Finally, for colonial monogamous bird species copulating during long time periods, acoustic signals are important for recognition within the pair (Mc Arthur, 1982). In addition, when copulating Griffon pairs emit loud cries it has been suggested that they act as signaling territorial occupation of the nest (Robertson, 1986; Negro & Grande, 2001), though this copulatory activity may also act as a warning (Margalida & Bertran, 2010).

In conclusion, contrary to expectations our findings show that nesting sites are scenarios with low frequencies of EPC attempts due to: (i) low levels of territorial intrusions which, when they occur, are not necessarily associated with sperm competition; (ii) the aggressive response of pairs facing territorial intrusions; (iii) the relatively high presence of pairs in the nests (on average 47.7%) during the presumed fertile period. However, the copulatory behavior of this species in the fertile period (increased frequency and duration of copulations) suggests that under pressure from the colonial breeding system (proximity of conspecifics and/or males that may not always stay close to their mates), a higher rate of copulation is the best preventive mechanism against relative uncertainty of paternity. On the other hand, the results suggest that males seem to make EPC attempts in sites far from the nest, coinciding with the collection of nest material. Under these circumstances, an increase in copulation attempts during the fertile period suggest that, for males, this is probably the most effective way to obtain the last copulations with their mates before egg-laying (Birkhead & Møller, 1992) and also to ensure their paternity. Griffon Vulture males play all or nothing on investment for a single egg and chick per breeding attempt, where besides their contribution in parental care is indispensable. In consequence, paternity loss would be too expensive for Griffon Vulture males.

Supplemental Information

Data S1 Dataset of copulations by griffon vultures

Click here for additional data file.

We thank Karl Schulze-Hagen, Tim R. Birkhead and two anonymous reviewers for their comments on improving the manuscript and Brian Hillcoat for revision of the English style.

Additional Information and Declarations

Competing Interests

Author Contributions

Animal Ethics

Data Availability

The authors declare there are no competing interests.

Joan Bertran conceived and designed the experiments, performed the experiments, wrote the paper, reviewed drafts of the paper.

Francesc Xavier Macià performed the experiments, reviewed drafts of the paper.

Antoni Margalida conceived and designed the experiments, performed the experiments, analyzed the data, contributed reagents/materials/analysis tools, wrote the paper, prepared figures and/or tables, reviewed drafts of the paper.

The following information was supplied relating to ethical approvals (i.e., approving body and any reference numbers):

All procedures regarding observational field study (Ref. 4925-2009/2011) were conducted according to the relevant Spanish legislation and following the conditions and guidelines approved by the Department of Agriculture, Livestock and Fisheries of the Government of Catalonia.

The following information was supplied regarding data availability:

Data can be found in Data S1.

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
