# Peer review of "How do colonial Eurasian Griffon Vultures prevent extra-pair mating?"

_PeerJ, doi:10.7717/peerj.1749_

## Round 0.1 · original submission · Major Revisions

· Academic Editor

Major Revisions

Dear authors

Thank you for submitting your manuscript to our journal. As you see our reviewers suggest a major revision of your ms. If you are willing to do so, we would be happy to reconsider your revised manuscript.

Michael Wink

Reviewer 1 ·

Basic reporting

The paper by Bertran et al. uses observational data from several Griffon vulture colonies in order to determine the mechanisms colony breeding griffon vultures use in order to prevent extra-pair copulations (EPC). The authors claim that their study adds information to the body of information of male behavior aimed at the prevention of EPC and cuckoldry. The paper is mostly written in clear English and well structured. In general the paper might benefit from some condensation as the findings principally confirms behavior that has previously been described in other avian species. The figure 1 to 4 could be drawn together to form two figures, or else should be posed together into a panel. The Excel data file lacks details like units (duration in seconds/minutes or??) in the headings. "Presence one" means time/frequency.....??? etc.

Experimental design

If the authors obtained a specific permit for their field work this should be stated with its corresponding reference.
I am concerned with a change in data collection throughout the four study years. In the material and methods sections the authors state that in 2010 they were able to discern if copulation attempts were successful. How were copulations attempts classified in the other years, How many of the observations were from which period?

Validity of the findings

As the authors state that none of the individuals was marked, I think the criteria for selection of WPC are correct and thus the results are sound. However I am concerned about the EPC events. Could the authors expand on how they are sure about the individuals involved?

Additional comments

In addition or conclusión f the aforementioned,
1- The paper should be condensed .
2- The English language, especially in the discussion section needs some revision, as an example:“Griffon Vulture males play all or nothing on investment for a single egg and chick per breeding attempt, where besides its contribution in parental care is indispensable.” Should read “Griffon Vulture males play all or nothing on investment for a single egg and chick per breeding attempt, where besides their contribution in parental care is indispensable.”
While the last sentence: “Consequently for them would mean losses paternity too expensive.” Makes no sense at all, although it was probably meant to mean: “In consequence, paternity loss would be too expensive for griffon vulture males.”

Reviewer 2 ·

Basic reporting

There are several problems:

The individual birds are not marked and could not be distinguished by other means. Even males and females cannot be distinguished.

Lines 254-255: The authors state that other vulture species do not feed their partners (courtship feeding). What about this species? Foraging must take quite a bit of time for Griffon Vultures. How much time are both sexes away from their nest? Are they foraging at different times? How long time is one partner alone at the nest? Is there a difference between both sexes? The authors just mention nest material collection, nothing about foreaging times.

The conclusions of the paper are based on two different time periods, the pre-fertile (PF) and fertile (F) (Line 169-172). Than the authors say that the extent of the fertile period is unknown. Finally the authors assume a fertile period of less than 14 days before egg laying. This makes the whole result uncertain. Are there no other references which may help to come to a conlusion, not just to guess the length?

Lines 214-220: EPCs were observed 20 - 200 m away from the nest site. Can it be excluded that this also occured much further away? How far was it possible to observe the birds? How far was it possible to hear the copulation calls? What was the distance to the nest furthest away from those observed? Would the authors have been able to hear or see EPCs there?

Experimental design

Ideally the birds would have been marked and it would have been possible to recognize them individually. Are there no colonies where at least some birds are marked?

The authors should write about the size of the colonies (max. distance from one side to the other of the colonies) to give an idea what activities were possible to observe from the observation points.

Validity of the findings

The validity is limited because of the problems described under "Basic reporting".

Additional comments

Apart from the problems mentioned under "Basic reporting" the text is too long and could be condensed.

It should also be mentioned how other authors overcame the problems mentioned in this and other species of vultures.

---

## Round 0.2 · Minor Revisions

· Academic Editor

Minor Revisions

Dear authors

Thanks for your careful revision. As you see one reviewer wants some minor revision. If you are doing this according to the suggestions, we are sure that we can soon accept your ms.

Thanks for submitting your work to PeerJ.

Greetings
Michael Wink

Reviewer 1 ·

Basic reporting

The authors of the paper “How do colonial griffon vultures prevent extra pair mating?” have condensed the paper and answered all of the reviewers concerns with view to formal aspects. They have added explanations in order to answer the concerns related to the contents, and thus clarified many of the aspects related to basic reporting within the paper.
With view to the study I am still somewhat concerned about the data collected and presented on EPC. As individuals were not marked and even being relatively small colonies I am not convinced about the accuracy of the data on EPC that occurred (presumably) away from the nest. Recording and analysis of the behavior at the nest (WPC, intrusions, etc.) are the core of this paper and it might be enough to report that EPC at the nest were not observed instead of including potentially incomplete data on EPCs away from the nest site that cannot be evaluated/analyzed objectively.
The text has been correctly amended with view to English expression.

Experimental design

As said above, in the present version the study design and data recording appear sound with view to the nest of the study pairs. For clarity I would suggest excluding data on supposed EPCs (other than on the nest site) by the studied pairs.

Validity of the findings

The findings on changes in behavior at the nest between the presumed pre-fertile and fertile period are sound and of interest. As no constant, sound and extensive information exists on EPC events away from the studied nests, conclusions on the effectiveness of the observed behaviors may be speculative, and should invite future studies on EPC behavior preferably in marked individuals.

Additional comments

As said above unclear information should probably omitted. Some more care should be employed in data presentation:
In the raw data table "percentage" should be replaced by "%", Also the figure panels present information clearer now, but lengeds should be placed on the axis instead of in the title As an example Figure 1: the upper graph should have copulation attempts/hour on the y axis and say "period" below fertile and pre-fertile. The title should be omitted as information is in the caption. Likewise the lower graph should have duration (sec) on the y-axis and "period" below the x-axis as mentioned before.

---

## Round 0.3 · accepted · Accept

· Academic Editor

Accept

Dear authors

Thanks for the revision. Congratulations! The ms is now fine and is accepted.

Greetings

Michael Wink